# DOLAVI Real-Life Study of Dolutegravir Plus Lamivudine in Naive HIV-1 Patients (48 Weeks)

**DOI:** 10.3390/v14030524

**Published:** 2022-03-04

**Authors:** Carmen Hidalgo-Tenorio, Juan Pasquau, David Vinuesa, Sergio Ferra, Alberto Terrón, Isabel SanJoaquín, Antoni Payeras, Onofre Juan Martínez, Miguel Ángel López-Ruz, Mohamed Omar, Javier de la Torre-Lima, Ana López-Lirola, Jesús Palomares, José Ramón Blanco, Marta Montero, Coral García-Vallecillos

**Affiliations:** 1Unit of Infectious Diseases, Virgen de las Nieves University Hospital, 18014 Granada, Spain; jpasquau@gmail.com (J.P.); malruz@ugr.es (M.Á.L.-R.); gvcoral@gmail.com (C.G.-V.); 2Unit of Infectious Diseases, University Hospital San Cecilio, 18016 Granada, Spain; vinudavg@gmail.com; 3Unit of Infectious Diseases, Torrecárdenas University Hospital, 04009 Almería, Spain; serfemu@gmail.com; 4Unit of Infectious Diseases, Hospital de Jerez, 11407 Jerez de la Frontera, Spain; alberto.jaterron@hotmail.com; 5Unit of Infectious Diseases, Hospital Lozano Blesa, 50009 Zaragoza, Spain; isanjoaquin@salud.aragon.es; 6Internal Medicine Service, Hospital Son Llatzer, 07198 Palma, Spain; apayeras@hsll.es; 7Unit of Infectious Diseases, Hospital Santa Lucia, 30300 Cartagena, Spain; onofrejmartinez@hotmail.com; 8Unit of Infectious Diseases, Hospital Complex of Jaén, 23007 Jaén, Spain; omarampa@gmail.com; 9Department of Internal Medicine, Hospital Costa del Sol, 29603 Marbella, Spain; jtorrel@gmail.com; 10Unit of Infectious Diseases, University Hospital Canarias, 38320 San Cristóbal de La Laguna, Spain; lopezlirola@hotmail.com; 11Internal Medicine Service, Hospital Santa Ana, 18600 Motril, Spain; jpalomaresr@hotmail.com; 12Unit of Infectious Diseases, Hospital San Pedro, 26006 Logroño, Spain; jrblanco@riojasalud.es; 13Service of Infectious Diseases, Hospital de La Fe, 46026 València, Spain; martamontero72@gmail.com

**Keywords:** DOLAVI, dolutegravir, lamivudine, real-world data, HIV

## Abstract

Brief: Real-world data in naïve HIV-1 patients demonstrate that dolutegravir plus lamivudine in a multiple tablet regimen is effective, safe, and satisfactory; it causes moderately increasing weight and abdominal circumference and is administrable on a test-and-treat strategy. Background: Our objectives were to determine the real-life effectiveness and safety of DT with dolutegravir (50 mg/QD) plus lamivudine (300 mg/QD) in a multiple-tablet regimen (MTR) in naïve PLHIV followed up for 48 weeks and to evaluate the compliance and satisfaction of patients. Material and methods: An open, single-arm, multicenter, non-randomized clinical trial from May 2019 through September 2020 with a 48-week follow-up. Results: The study included 88 PLHIV patients (87.5% male) with a mean age of 35.9 years; 76.1% were MSM patients. The mean baseline CD4 was 516.4 cells/uL, with a viral load (VL) of 4.49 log_10_, and 11.4% were in the AIDS stage. DT started within 7 days of first specialist consultation in all patients and the same day in 84.1%; 3.4% had baseline resistance mutations (K103N, V106I + E138A, and V108I); 12.5% were lost to follow-up. At week 48, 86.3% had VL < 50 cop/uL by intention-to-treat analysis and 98.7% by per-protocol (PP) analysis. Virological failure (VF) was recorded in 1.1%, with no resistance mutation. One blip was detected in 5.2% without VF. Three reported anxiety, dizziness, and cephalgia, respectively, at week 4 and one reported insomnia at week 24; none reported adverse events at week 48. The mean weight was 4 kg higher at 48 weeks (*p* = 0.0001) and abdominal circumference 3 cm larger at 24 weeks (*p* = 0.022). No forgetfulness occurred in 98.7% of patients. Patient satisfaction was 90/100 at 4, 24, and 48 weeks. Conclusion: Real-world data demonstrate that dolutegravir plus lamivudine in MTR is effective, safe, and satisfactory, moderately increasing weight and abdominal circumference and administrable on a test-and-treat strategy.

## 1. Introduction

People living with HIV (PLHIV) have experienced an exponential increase in survival and life expectancy over the past two decades thanks to antiretroviral treatment (ART). The resulting improvement in morbidity and mortality has to HIV becoming a chronic infection [1]. Life-long ART is currently recommended in the absence of a definitive cure; however, although the effectiveness and safety of antiretroviral drugs (ARVs) have gradually improved, they are not yet free of toxicity [2]. Therefore, research efforts focused on simplifying the treatment and reducing the number of ARVs needed to control HIV infection [3]. Proposed strategies include monotherapy with protease inhibitors (PI), mainly in undetectable treatment-experienced patients [4], which is not currently prescribed in clinical practice. Dual therapies (DTs) were recently studied in naïve or pretreated patients, evaluating combinations of dolutegravir with lamivudine (3TC) [5,6], dolutegravir with rilpivirine [7,8], and rilpivirine with darunavir/ritonavir [9].

Real-life studies [10,11,12] and clinical trials [13,14] reported that DT with dolutegravir and lamivudine has an effectiveness of close to 100% in patients with experience of ART and previous virological suppression. In naïve HIV patients, two clinical trials (GEMINI 1 and 2) reported a virological success rate of ≥90% at week 48 with DT [15] and at week 96 [16], confirming the non-inferiority of the DT. However, although clinical trials are the gold standard for the approval of novel treatments, the ideal conditions in which they are conducted, with a careful selection of patients, may not reflect those in the actual clinical setting. The publication of so-called “real-world data” can help bridge the gap between clinical trials and real-life practice, increasing the clinical relevance of findings and supporting the incorporation of novel therapies and technologies into routine clinical practice [17].

Hence, the main objective of this real-life study of naïve HIV patients was to determine the effectiveness of DT with dolutegravir 50 mg/QD plus lamivudine 300 mg/QD in a multiple-tablet regimen (MTR). Our secondary objectives were: to describe the type of patient prescribed with this biotherapy, to measure the interval between first specialist consultation and DT initiation (deferred or immediate/early treatment); to evaluate its possible application in a “test-and-treat” approach; and to determine the following: gradient of the decrease in viral load (VL), increase in CD4 lymphocyte levels, type of virological failure (VF), the influence of previous mutations detected in baseline genotypic resistance tests (GRTs), the safety of the treatment, and the satisfaction of patients and their retention within the health system.

## 2. Patients and Methods

STUDY DESIGN This prospective, longitudinal, post-authorization, single-arm, multicenter study consecutively included naïve HIV patients starting ART with dolutegravir 50 mg/QD and lamivudine 300 mg/QD in MTR, from May 2019 through September 2020 with a 48-week follow-up.

PATIENTS: Inclusion criteria: The presence of HIV infection, age > 17 years, prescription by an attending physician of dolutegravir 50 mg/QD and lamivudine 300 mg/QD in MTR, and written informed consent to participate in the study.

Exclusion criteria: Pregnancy, non-utilization of contraception in women of childbearing age, life expectancy less than the study period, the expectation by attending physician of treatment change before the study ends (not due to adverse drug effects but rather to incompatibility with other drugs to be administered to the patient), and protocol violation.

Origin: The patients came from the infectious disease or internal medicine departments of 13 public hospitals in Spain.

### 2.1. Intervention

Epidemiological, clinical, and analytical data were gathered in accordance with national data protection legislation (Organic Law 3/2018, 5 December). The study complied with the principles of the Helsinki Declaration and was approved by the clinical research ethics committees of the Virgen de las Nieves University Hospital of Granada (in accordance with decree 8/2020, 30 January, which regulates the care and biomedical research ethics in Andalusia) and all other participating centers.

At their baseline visit, patients were informed about the study conditions and objectives and were asked for their written consent. At the same visit, data were gathered on anthropometric variables (height, weight, and abdominal circumference), and blood samples were drawn for: blood count, biochemistry (creatinine, urea, CKD-EPI, GOT, GPT, GGT, FA, calcium, phosphorus, total cholesterol [TC], HDL, LDL, and TC:HDL ratio), CD4 and CD8 lymphocyte counts, CD4/CD8 ratio, HIV VL, GRT, and serology studies of hepatitis B virus (HBV), hepatitis A virus (HAV), hepatitis C virus (HCV), syphilis, cytomegalovirus, and toxoplasma.

Data were again collected on the above anthropometric measures and blood analysis results (except for serology) during follow-up visits at weeks 4, 24, and 48, when they were also asked to report any adverse effects and to complete EQ-5D [18] and SMA-Q [19] questionnaires on their satisfaction and compliance with the treatment, respectively. GRT was requested when VF was confirmed. Retention within the health care system was evaluated in terms of losses to the follow-up.

### 2.2. Definition of Variables

Effectiveness: Plasma VL < 50 cop/mL at week 48 according to intention-to-treat (ITT; FDA Snapshot algorithm) and per-protocol (PP) analyses.

Virological failure: Two consecutive VLs > 50 cop/mL after previously achieving non-detectability during the DT.

Blip: Viral load > 50 cop/mL and <200 cop/mL after achieving undetectable VL, which is again undetectable in a test within 15 days.

Deferred treatment: Initiation at ≥2 weeks after the initial specialist consultation.

Test and treat strategy: Either immediate treatment on the day of the initial consultation (based on single blood count and biochemistry analysis with serology result confirming HIV) or early treatment, i.e., within two weeks of the initial consultation.

ART dropout: Change from the DT to another ART requested by patients due to adverse effects or to their preference for a single-tablet regimen (STR), among other reasons. Losses to the follow-up are not considered in this calculation.

Retention in the healthcare system: Defined by the percentage of patients who were not lost to the 48-week follow-up.

Classification of weight as a function of BMI (kg/m^2^) [20]: Underweight: <18.5; Normal weight: 18.5–24.9; Overweight: 25–29.9; Obesity Class 1: 30–34.9, Class 2: 35–39.9; and Class 3: ≥40.

### 2.3. Classification of Adverse Events

Mild: no antidote or treatment needed; brief hospitalization. Moderate: treatment modification required (e.g., change in dosage, the addition of another drug), but the DT is not discontinued; a longer stay or the addition of a specific treatment may be necessary. Severe: The adverse drug reaction is life-threatening and requires the withdrawal of the DT and the initiation of a specific treatment. Fatal: The adverse reaction directly or indirectly contributes to the patient’s death.

Self-perceived quality of life: This was evaluated by the patients using the visual analog scale (VAS) of the EuroQol-5D (EQ-5D) questionnaire (score range 0–100). Results were expressed as percentages [18].

Adherence: The simplified medication adherence questionnaire (SMAQ) was self-completed by the patients to evaluate their adherence to the ART protocol [19].

Sample size: The sample size estimation assumed effectiveness of 92% and considered a bilateral confidence interval of 95% and an error of 10%. A final sample size of 88 individuals was selected to allow for a possible loss to follow-up of 10% (Table 1).

Statistical methods: In a descriptive analysis, means with standard deviations were calculated for quantitative variables that were normally distributed (by the Kolmogorov–Smirnov test), medians and percentiles for those that were not, and absolute frequencies for qualitative variables. DT effectiveness was confirmed using the FDA Snapshot algorithm, defining virological success as plasma VL < 50 cop/mL at week 48 by ITT analysis and considering VF, treatment discontinuation, or loss to the follow-up as failures. The effectiveness at week 48 was also calculated according to PP analysis, considering VF alone as a failure. IBM SPSS Statistics for Windows version 25.0 (IBM Corp, Armon, NY, USA) was used for data analyses.

The study was recorded in ClinicalTrials.gov. Identifier: NCT04002323.

## 3. Results

### 3.1. Study Population

The study included 88 people living with HIV (PLHIV). The mean age was 35.9 years; 87.5% were men, and 76.1% were men who have sex with men (MSM). The mean baseline CD4 was 516.4 cells/uL (8% had Cd4 < 200 cells/uL), and the mean VL was 4.49 log_10_ (20% had VL > 100,000 cop/mL); 11.4% of patients were in the AIDS stage, and 1.1% had positive HBV surface and core antibodies. DT was initiated within one week of the initial specialist consultation in 100% of patients (median of 0 days: range, 0–7 days) and on the same day in 84.1%. Baseline resistance mutations (K103N, V106I + E138A, and V108I, respectively) were detected in three patients (3.4%). Table 1 list the remaining results. Eleven patients (12.5%) were lost to the follow-up, as shown in Figure 1. 

### 3.2. Effectiveness

At week 48, 86.3% of patients had VL < 50 cop/uL by ITT analysis (FDA snapshot algorithm) and 98.7% by PP analysis. The follow-up was completed by 14 of the 17 patients with baseline VL > 100,000 cop/mL, and all of them had < 50 cop/mL at week 48. One patient (1.1%) had VF at week 48 due to poor adherence, with no GRT-detected mutations. He was then prescribed tenofovir-alafenamide/emtricitabine/bictegravir, although non-detectability was not achieved, and he had plasma VLs of 215 cop/uL and 183 cop/mL during the 24 weeks after its initiation; this patient showed no mutations in a new GRT. 

His ART was again changed, to tenofovir-alafenamide/emtricitabine/darunavir/cobicistat, showing VLs of 311 and 211 cop/mL during the 24 weeks of this treatment (Table 2).

During the 48-week follow-up, four patients (5.2%) had one blip, with plasma VLs of 83, 93, 97, and 109 cop/mL, respectively; none of these patients presented with VF.

### 3.3. Blood Analysis

The mean CD4 count was significantly increased at week 4 (516.4 ± 267.2 cells/uL baseline vs. 675.9 ± 336.1 cells/uL week 4; *p* = 0.0001) and continued to rise, as shown in Figure 2; this count was 305 cells/uL higher at week 48 than at baseline (821.8 vs. 441.6, *p* = 0.0001). The CD4/CD8 ratio also increased from baseline to week 48 (0.60 ± 0.37 vs. 0.93 ± 0.54, *p* = 0.0001). VL was undetectable in 70.2% of patients at week 4. The creatine level increased by only 0.1 mg/dL (0.8 mg/dL vs. 0.9 mg/dL; *p* = 0.0001) from baseline to week 48, with a reduction in creatine clearance of only 9 mL/min (109 vs. 100 mL/min.; *p* = 0.0001). The mean TC increased by 19 mg/dL (158.4–177.2 mg/dL; *p* = 0.0001) from baseline to week 48, whereas there was no significant change in TC/HDL ratio (3.95 ± 1.3 vs. 4.05 ± 1.25, *p* = 0.579) or triglyceride count (112.2 vs. 122.8 mg/dL; *p* = 0.25) during the study period, (Table 3).

### 3.4. Anthropometrics

The mean weight of patients increased by 4 kg over the 48-week follow-up period (72.3 vs. 76.2 Kg; *p* = 0.0001) and their mean BMI increased from 23.8 ± 3.9 to 25.2 ± 4.43 (*p* = 0.0001), i.e., from normal weight at baseline to overweight at week 48. The distribution of the BMI-based classification of patients over the study period was: low weight in 3.5% at baseline vs.2.9% at week 48, *p* = 0.56; normal weight in 61.6% vs. 56.5%, respectively, *p* = 1; overweight in 29.1% vs 26.1%, respectively, *p* = 0.58; and obesity in 5.8% vs. 14.5%, respectively, *p* = 0.06 (Figure 3). Waist circumference was increased by 3 cm at 24 weeks of follow-up (*p* = 0.022).

### 3.5. Adverse Effects

Anxiety, dizziness, and cephalgia were described by one patient each at week 4, insomnia was reported by one patient at week 24 and no adverse event by any patient at week 48.

### 3.6. Adherence and Satisfaction

All patients evidenced 100% adherence to the treatment in the PP analysis, no follow-up visits were missed, and no forgetfulness was observed. The EQ-5D VAS-measured satisfaction level was 90 (IQR: 80–95) at week 4, 90 (IQR: 80–100) at week 24, and 90 (85–100) at week 48. Eleven patients (12.5%) were considered lost to the health care system.

## 4. Discussion

The patients in this study were predominantly young MSM with good viral and immune status, and only 11.4% had AIDS. The DT was administered within one week of the first specialist consultancy in 100% of the patients and on the same day in >80%.

At the end of the 48-week follow-up, 86.3% of participants had a plasma VL < 50 cop/mL by ITT analysis and 98.7% by PP analysis. Only one patient presented with VF attributed to poor adherence; no resistance mutations were detected in this patient, likely due to the presence of moderate-low level viremia. VF was not observed in the four patients who experienced a blip. Frequent blips or plasma VL counts > 200 cop/mL are associated with a higher risk of VF and the emergence of multi-drug resistance [21,22]. This suggests that the patients with blips did not fail because their VL ranged from 83 cop/mL to 109 cop/mL, and each only had one blip during the follow-up period. Only two non-randomized, multicenter phase III clinical trials specifically evaluated the effectiveness of the rapid initiation of an ART. One of these, the DIAMOND trial [23], administered tenofovir-alafenamide/emtricitabine/darunavir/cobicistat (SYMPTUZA[c]) to 109 patients within the first two weeks of their diagnosis before receiving blood analysis results; and described effectiveness at week 48 of 84% by FDA Snapshot, and 96% by PP analysis, with no patients developing VF or genotypic resistance mutations. In the other trial, STAT [24], dolutegravir plus lamivudine (DOVATO (c)) were administered in STR to 131 patients; this regimen was modified in 8 patients during the first 24 weeks (5 for HBV infection, 1 for M184V mutation in baseline GRT, 1 for adverse effects, and 1 by patient request). Plasma VL was <50 cop/mL in 78% (FDA Snapshot) at week 24, and in 82% (FDA Snapshot) and 92% (PP analysis) at week 48. No patients developed VF with resistance mutation [24]. The STAT results were similar to the present findings.

The loss to follow up in the present study (12.5%) is also comparable with that recorded in the DIAMOND (11%) and STAT (14%) trials, despite conducting part of the enrolment and follow-up process during the SARS-CoV-2 pandemic (recruitment from May 2019 through September 2020 with 48-week follow-up), when some individuals avoided visits to the hospital for fear of contagion. This DT (dolutegravir plus lamivudine) therefore appears to be an optimal treatment to ensure that patients are retained within the health system, whether administered in STR or MTR. In addition, seven out of ten patients were non-detectable at week 4, one of the advantages of the test-and-treat strategy [25] and recommended in the new definition of therapeutic success in PLHIV, prioritizing the administration of an ART that rapidly reduces plasma VL, especially in at-risk individuals [26]. The rapid initiation of ART in naïve patients, even in those with mental, substance abuse, or homelessness problems, demonstrated enhanced individual and public health benefits in comparison to conventional care by reducing HIV transmission risk and improving treatment adherence and the retention of patients in the health care system [27]. This approach was proven effective both in nations with limited economic resources [28] and in high-income countries such as the USA [29].

The treatment produced no changes in the lipid profile of the patients, as previously observed in clinical trials of this DT in trials with naïve patients [15,16] and in real-life study [30]. In another study, lipid profile improvements were observed in patients who switched to this DT from an ART that contained a boosting agent [31].

The mean weight classification of the present patients changed from normal weight to overweight over the 48-week follow-up, and their waist circumference increased by 3 cm over the first 24 weeks. Most clinical trials and observational and cohort studies of integrase inhibitors as first-line ART regimens have described a weight gain in naïve HIV patients. Dolutegravir is the drug most frequently associated with this increase; however, the underlying mechanism was not established, and it is not known whether it is related to lipohypertrophy with visceral fat deposits [32].

Despite the administration of a specific questionnaire on potential adverse events, these were very rare. In addition, the patients expressed a high level of satisfaction with the treatment, which was fully adhered to by patients who completed the follow-up. In the DIAMOND trial [21], only one patient experienced an adverse effect (rash) that prompted treatment modification, and very high scores (close to the maximum) were obtained for patient satisfaction. Likewise, treatment was changed due to an adverse effect in only one patient in the STAT trial [22].

The present study is limited by its open, non-randomized design, similar to the only two trials that analyzed early ARV initiation in naïve HIV patients. However, it offers a prospective longitudinal analysis of the real-life effectiveness of this DT when prescribed by HIV specialists, contributing real-world data.

In conclusion, DT with dolutegravir and lamivudine in MTR is an effective, safe, and satisfactory strategy for rapid or immediate initiation and achieves a high rate of retention within the health care system. This DT increases the weight and BMI of patients who globally pass from normal weight to overweight within 48 weeks and already have a significantly expanded abdominal circumference at 24 weeks.

## Figures and Tables

**Figure 1 viruses-14-00524-f001:**
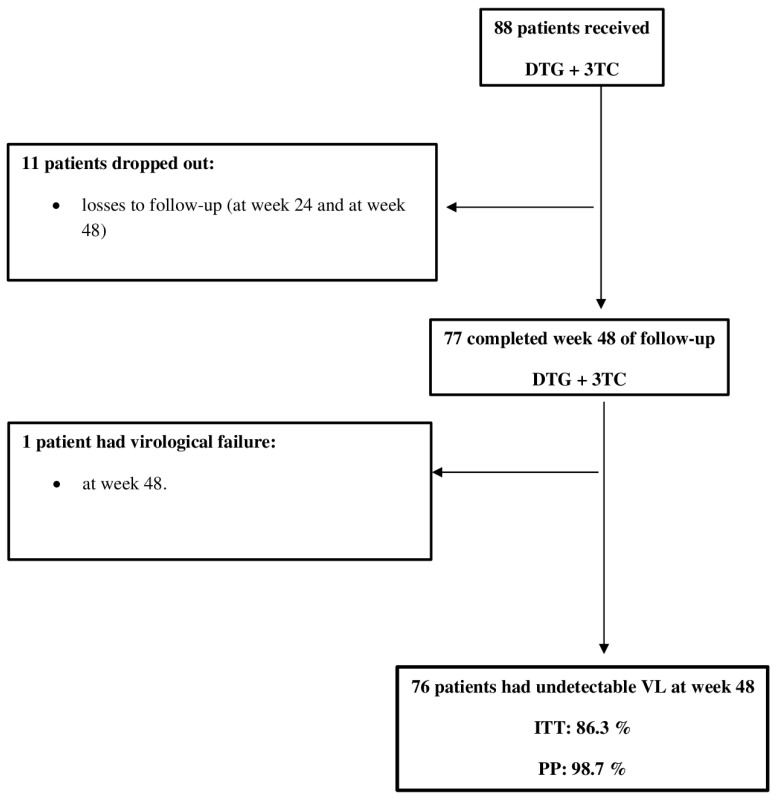
Flowchart of patients treated with dolutegravir (DTG) plus lamivudine.

**Figure 2 viruses-14-00524-f002:**
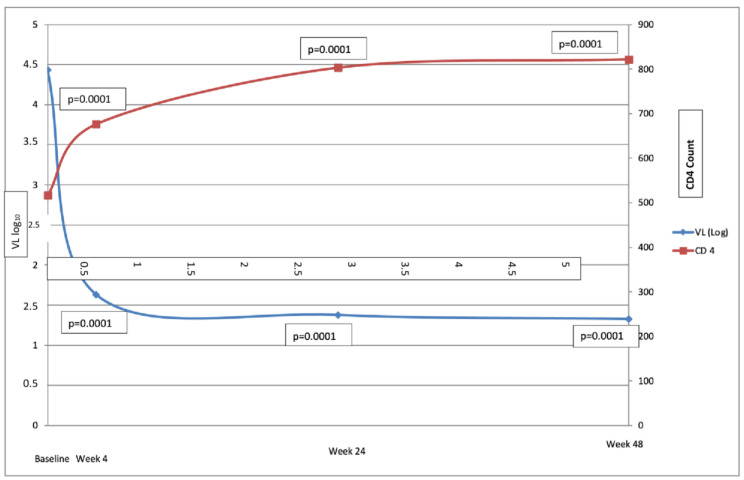
Time course of CD4 and viral load (baseline and weeks 4, 24, and 48 weeks).

**Figure 3 viruses-14-00524-f003:**
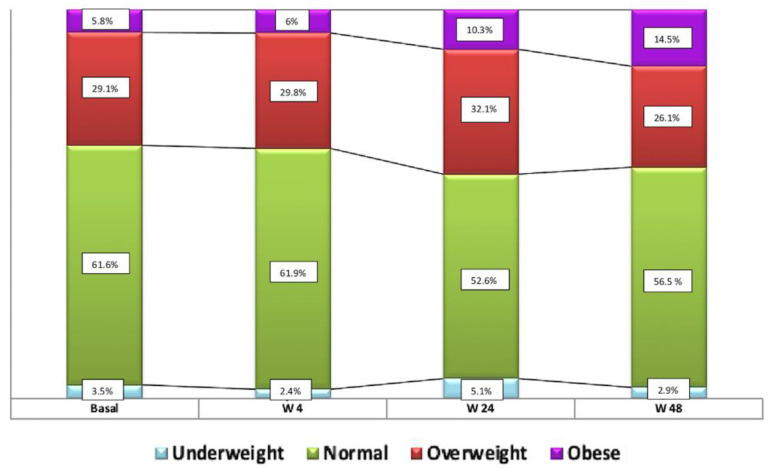
Distribution of patient weight classification by BMI at baseline and weeks 4, 24, and 48.

**Table 1 viruses-14-00524-t001:** General description of the population.

Variable	N = 88
**Age (year)**, mean (±SD)	35.8 (±12.2)
**Male, n (%)**	77 (87.5)
**Rapidly initiated DT, median (IQR)**	0 (0–7)
**Baseline HIV Viral Load, log_10_**, median (IQR)	4.49 (4–4.9)
Viral Load > 100,000 cop/mL, n (%)	17 (20)
**Resistance mutations baseline, n (%)**	3 (3.4)
K103 N	1 (33.3)
V106I, E138A	1 (33.3)
V108I	1 (33.3)
**Baseline CD4, (Cell/uL)**, mean (±SD)	512.2 ± 250.3
CD4 < 200 cell/uL, n (%)	7 (8)
**Baseline CD4/CD8 ratio**, mean (±SD)	0.6 (±0.37)
**CDC AIDS stage (A3, B3, C), n (%)**	10 (11.4)
A1	42 (47.7)
A2	33 (37.5)
A3	8 (9.1)
B1	1 (1.1)
B2	2 (2.3)
C2	1 (1.1)
C3	1 (1.1)
**Chronic hepatitis C cured, n (%)**	3 (3.4)
Positive HVB surface and core antibodies, **n (%)**	1 (1.1)
**Risk factor for HIV infection, n (%)**	
Heterosexual	18 (20.5)
MSM	67 (76.1)
IVDU	3 (3.4)
**Smoker, n (%)**	47 (53.4)
**Non-alcohol consumer, n (%)**	62 (70.5)
**Social drinker, n (%)**	26 (29.5)
**Weight, mean** (±SD) (Kg)	72.3 (11.8)
**BMI, mean** (±SD), Kg/m^2^	23.8 (3.9)
**Waist, mean** (±SD) (cm)	84.7 (12.7)
**Underweight, n (%)**	3 (3.5)
**Normal weight, n (%)**	53 (61.6)
**Overweight, n (%)**	25 (29.1)
**Obese**	5 (5.8)

**Table 2 viruses-14-00524-t002:** Virological failures.

Patient	Age	Sex	Baseline VL	Baseline CD4	VL in VF	Baseline RS **	RS of VF	Week of * VF	Observations
**1**	48	M	370,000 cop/uL	235 cell/uL	1st VL: 423 cop/uL 2nd VL: 763 cop/uL	No mutations	No mutations	24	Poor adherence After starting ART 1° BIC/FTC/TAF, with VL 215 and 183 cop/mL RS: negative 2° DRV/cob/FTC/TAF with VL 311 and 2211 cop/mL RS: Negative

M: male; ART: antiretroviral therapy; 3TC: Lamivudine BIC: bictegravir, FTC: emtricitabine; TAF: tenofovir alafenamide; DRV/cob: darunavir/cobicistat; VL: HIV viral load; * VF: virological failure; ** RS: genotypic resistance test.

**Table 3 viruses-14-00524-t003:** Analytical changes between baseline and weeks 4, 24, and 48.

	Baseline	Week 4	Week 24	Week 48	*p*-Value (Baseline vs. Week 48)
Creatinine (mg/dL), mean ± SD Creatinine Clearance (CKD-EPI), mL/min, mean ± SD	0.82 ± 0.1 109 ± 16	0.94 ± 0.14 99.5 ± 14.2	0.94 ± 0.15 99.3 ± 16.1	0.9 ± 1.04 100.8 ± 15.7	0.0001 0.0001
Total cholesterol (mg/dL), mean ± SD	158.4 ± 39.8	177.1 ± 39.5	174.8 ± 35.7	177.2 ± 38.7	0.0001
HDL cholesterol (mg/dL), mean ± SD	48.3 ± 43	45.3 ± 11.1	46.2 ± 11.5	46.3 ± 12.3	0.005
LDL cholesterol (mg/dL), mean ± SD	102.7 ± 34.8	112 ± 35.2	110.5 ± 35.2	112.1 ± 32.3	0.012
TC/HDL ratio, mean ± SD	3.9 ± 1.3	4.2 ± 1.26	4.1 ± 1.1	4 ± 1.3	0.579
Triglycerides (mg/dL), mean ± SD	112.2 ± 66.4	121.8 ± 85.9	126.3 ± 76.7	122.8 ± 81	0.25

## Data Availability

Registration and protocol: ClinicalTrials.gov Identifier: NCT04002323.

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
