# Peer review of "DOLAVI Real-Life Study of Dolutegravir Plus Lamivudine in Naive HIV-1 Patients (48 Weeks)"

_viruses, 2022, doi:10.3390/v14030524_

Round 1
Reviewer 1 Report
This is an interesting real-life study on lamivudine-dolutegravir DT in naive HIV patients.
I would like to knows how many patients show HIV-RNA at baseline > 100.000 cp/mL. A subanalysis, focusing on these patients, should be presented.
There are some missing in Table 1, in particular the number of male, the IQR of baseline HIV VL, number of pts with VL > 100.000 cp/mL and the numbers of pts with CD4 < 200 cell.
I suggest to evaluate a comparison with the paper "Short Communication: Efficacy and Safety of Dolutegravir Plus Lamivudine as a First-Line Regimen in Clinical Practice. Cicculo et al. AIDS Res Hum Retroviruses. 2021 Jun;37(6):486-488. doi: 10.1089/AID.2020.0276. Epub 2021 Mar 19.PMID: 33587008"
Author Response
We are very grateful to the Editor and Reviewers for their insights and recommendations, which have all been taken into account in a thorough revision of our paper, as detailed in our point-by-point responses below
Responses to Reviewer 1
R1: This is an interesting real-life study on lamivudine-dolutegravir DT in naive HIV patients.
1-R1: I would like to know how many patients show HIV-RNA at baseline > 100.000 cp/mL. A subanalysis, focusing on these patients, should be presented.
Response: This information is provided in the revised Results section (highlighted in yellow):.
Results: 3.1. Study Population “The study included 88 people living with HIV (PLHIV). The mean age was 35.9 years; 91% were men, and 76.1% were men who have sex with men (MSM). The mean baseline CD4 was 516.4 cells/uL (7% had Cd4 <200 cells/uL), and the mean VL was 104,828 cop/mL (20% had VL> 100,000 cop/mL).
Response: We have added a sub-analysis on the effectiveness in patients with CV>100,000 cop/mL: The follow-up was completed by 14 of the 17 patients with baseline VL >100.000 cop/mL, and all of them had < 50 cop/mL at week 48 (lines 174-175)
2- R1: There are some missing in Table 1, in particular the number of males, the IQR of baseline HIV VL, number of pts with VL > 100.000 cp/mL and the numbers of pts with CD4 < 200 cell.
Response: We apologize for this unfortunate error. These data are now reported in Table 1 (highlighted in yellow).
3-R1: I suggest to evaluate a comparison with the paper "Short Communication: Efficacy and Safety of Dolutegravir Plus Lamivudine as a First-Line Regimen in Clinical Practice. Cicculo et al. AIDS Res Hum Retroviruses. 2021 Jun;37(6):486-488. doi: 10.1089/AID.2020.0276. Epub 2021 Mar 19.PMID: 33587008"
Response: We are grateful for this suggestion. We now cite this paper (our reference 30) in the revised Discussion (lines 269-270).

Reviewer 2 Report
The main objective of this real-life study of naïve HIV patients was to determine the effectiveness of DT with dolutegravir plus lamivudine in multiple-tablet regimen and then many others, perhaps too ambitious in relation to the sample size and the limited follow-up
it is a well designed work with a cohort that is not too large but sufficient to produce interesting results.
nevertheless many points need to be clarified before considering it for publication
Line 60
“… and at week 96 with triple therapy (16)”
please correct this sentence;
further, update the references with AIDS Jan 1, 2022
table 1
the absolute value of patients CD4< 200 cell/uL is lost
Line 169
“One patient (1.1%) had VF at week 48 due to poor adherence, with no GRT-detected mutations”
How was poor adherence assessed (only by questionnaire?) and when VF occurred in the intervals described in tab 2?
Line 180-181
At what time points were these blips observed? After achieving the NR?
Fig 2
there is an error in the HIV-RNA scale
At what time points was the LV measured? T0, w4, w24 and w48?
even in subjects with blip?
Line 218
Eleven patients (12.5%) were considered lost to the health care system
At what time were these patients lost? For what reason? What characteristics did they have and what partial responses did they demonstrate? The work refers only to patients who have concluded the observation, and instead these eleven arouse great interest
Line 225
“no resistance mutations were detected in this patient”
please specify the VL values corresponding to the determinations of the resistance tests; have they always been obtained on plasma, even with low copies?
Line 247
“one of the advantages of the test-and treat strategy (25) and recommended in the new definition of therapeutic success in PLHIV”.
But the numerous advantages of test and treat strategy, well described below, do not always include the rapid reduction of viremia, which could be observed even a few weeks after diagnosis, in a patient with chronic infection.
Furthermore, 12% of lost do not correspond to optimal retention in care even if they cannot be connected to the test and treat.
Author Response
We are very grateful to the Editor and Reviewers for their insights and recommendations, which have all been taken into account in a thorough revision of our paper, as detailed in our point-by-point responses below.
Responses to Reviewer 2:
The main objective of this real-life study of naïve HIV patients was to determine the effectiveness of DT with dolutegravir plus lamivudine in multiple-tablet regimen and then many others, perhaps too ambitious in relation to the sample size and the limited follow-up. It is a well designed work with a cohort that is not too large but sufficient to produce interesting results.
Nevertheless many points need to be clarified before considering it for publication
1- Line 60“… and at week 96 with triple therapy (16)” please correct this sentence;
Response: This sentence has been corrected (highlighted in yellow)
2- further, update the references with AIDS Jan 1, 2022
Response: This has been done, as follows:
16-Cahn P, Sierra Madero J, Arribas J, Antinori A, Ortiz R, Clarke R.et al. Three-year durable efficacy of dolutegravir plus lamivudine in antiretroviral therapy - naive adults with HIV-1 infection AIDS 2022; 36: 39-48.
3- table 1: the absolute value of patients CD4< 200 cell/uL is lost
Response: We apologize for this oversight. This value is now given
4- Line 169
“One patient (1.1%) had VF at week 48 due to poor adherence, with no GRT-detected mutations”
How was poor adherence assessed (only by questionnaire?) and when VF occurred in the intervals described in tab 2?
Response: The adherence was assessed not only by using the questionnaire described in Material and Methods but also by confirming whether the patient picked up the medication in accordance with the agreed regimen.
Table 2 now exhibits time intervals for VFs.
5- Line 180-181
At what time points were these blips observed? After achieving the NR?
Response: We have addressed this question by including the following definition of blips under the heading “Definition of variables” (lines 120-121):
Blip: Viral load > 50 cop/mL and < 200 cop/mL after achieving undetectable VL, which is again undetectable in a test within 15 days.
6- Fig 2
there is an error in the HIV-RNA scale
Response: This unfortunate error has been corrected.
At what time points was the LV measured? T0, w4, w24 and w48?
even in subjects with blip?
Response: These were the time points, even in patients with blips. When there was a suspected blip, the viral load was again measured after an interval of 15 days to test whether it was truly a blip (i.e., again undetectable) or virological failure.
In material y methods: 2.2 Definiton of variables: Blip: Viral load > 50 cop/mL and < 200 cop/mL after achieving undetectable VL, which is again undetectable in a test within 15 days.(line 120-121)
7- Line 218
Eleven patients (12.5%) were considered lost to the health care system. At what time were these patients lost? For what reason? What characteristics did they have and what partial responses did they demonstrate? The work refers only to patients who have concluded the observation, and instead these eleven arouse great interest
Response: The most likely explanation for losses is that the 48-week follow-up period, which ended in September 2021, coincided in part with the arrival of the COVID epidemic. This led many patients in all departments to miss follow-up visits and in general avoid the hospital for fear of contagion. This fact is now referred to in the revised Discussion (lines 253-256). Nevertheless, some patients considered lost to the system are now returning to see the specialist as fears subside.
8- Line 225
“no resistance mutations were detected in this patient”
please specify the VL values corresponding to the determinations of the resistance tests; have they always been obtained on plasma, even with low copies?
Response: Information on this patient's virologic failure is reported in Table 2, showing viral loads of 423 and 763 cop/uL. No resistance mutation was found
9- Line 247
“one of the advantages of the test-and treat strategy (25) and recommended in the new definition of therapeutic success in PLHIV”.
But the numerous advantages of test and treat strategy, well described below, do not always include the rapid reduction of viremia, which could be observed even a few weeks after diagnosis, in a patient with chronic infection.
Furthermore, 12% of lost do not correspond to optimal retention in care even if they cannot be connected to the test and treat.
Response: As reported by Ford N et al (25) in their meta-analysis, one advantage of test-and-treat is the rapid drop in viral load. In our study, 70.2% of patients had VL<50 cop/uL at week 4, which not only benefits the patient but also reduces the risk of transmission, assisting in the control of the AIDS pandemic alongside other approaches (PreP, condom use, etc.). As noted above (see response to point 7) and in the revised text, despite the arrival of the COVID pandemic, the loss to follow-up was similar to that in previously published test-and-treat studies (DIAMOND and STAT).
